# ER–Mitochondria Contact Sites Reporters: Strengths and Weaknesses of the Available Approaches

**DOI:** 10.3390/ijms21218157

**Published:** 2020-10-31

**Authors:** Flavia Giamogante, Lucia Barazzuol, Marisa Brini, Tito Calì

**Affiliations:** 1Department of Biomedical Sciences, University of Padova, 35131 Padova, Italy; flavia.giamogante@unipd.it (F.G.); lucia.barazzuol@studenti.unipd.it (L.B.); 2Department of Biology, University of Padova, 35131 Padova, Italy; 3Padova Neuroscience Center (PNC), University of Padova, 35131 Padova, Italy

**Keywords:** ER–mitochondria tethering, split-GFP, SPLICS, organelle contact sites

## Abstract

Organelle intercommunication represents a wide area of interest. Over the last few decades, increasing evidence has highlighted the importance of organelle contact sites in many biological processes including Ca^2+^ signaling, lipid biosynthesis, apoptosis, and autophagy but also their involvement in pathological conditions. ER–mitochondria tethering is one of the most investigated inter-organelle communications and it is differently modulated in response to several cellular conditions including, but not limited to, starvation, Endoplasmic Reticulum (ER) stress, and mitochondrial shape modifications. Despite many studies aiming to understand their functions and how they are perturbed under different conditions, approaches to assess organelle proximity are still limited. Indeed, better visualization and characterization of contact sites remain a fascinating challenge. The aim of this review is to summarize strengths and weaknesses of the available methods to detect and quantify contact sites, with a main focus on ER–mitochondria tethering.

## 1. Introduction

In eukaryotes, organelles are delimited by a membrane that enables them to perform specific and unique tasks, whose characterization has been the goal of researchers for many decades. In the last years, however, the inter-organelle communication received momentum and represents an emerging aspect in cell biology. It is now well established that organelles are not isolated, but they are interconnected, thereby forming a communicating network [1]. Proteins identified at the interface between organelles seem to act as tethers that physically bridge two opposing membranes and join them to each other [2,3]; others have a specific role that occurs at the contact sites, such as the ability to transfer small molecules in a non-vesicular manner [4,5].

The interplay between the endoplasmic reticulum (ER) and mitochondria has been one of the first identified and, to date, still represents one of the best characterized forms. ER–mitochondria juxtaposition has been firstly observed by electron microscopy (EM) in the late 1950s in rat tissue; however, at that time, it was considered an artifact due to fixation [6]. Conversely, later approaches compatible with living cells analysis have validated their existence [7]. Now, it is well established that ER–mitochondrial tethering plays a crucial role in several cellular pathways, such as Ca^2+^ homeostasis [8,9], lipid transfer [10], mitochondrial dynamics [11], apoptosis, and mitophagy [12,13]. Many shreds of evidence report that its alteration is linked to the development of different disorders including diabetes [14], cancer [15], and neurodegenerative diseases (e.g., Alzheimer’s disease (AD), Parkinson’s disease (PD), and amyotrophic lateral sclerosis (ALS) [16]). 

However, most of the research aimed at characterizing ER–mitochondria contact-site components and their functions face the complex issue of unequivocally identifying the contact, defining its function, and monitoring its changes upon perturbations. To date, several methods need to be combined to extract all the mentioned information since an optimized approach that can provide them at once is still missing. Indeed, the transient nature and variable abundance of ER–mitochondria interactions within different cell types make the development of suitable detecting methods a big challenge. During the past decades, considerable effort has been spent to ameliorate the available methods and develop new ones, leading to the possibility of further investigate on ER–mitochondria contact sites and provide more information. Starting from this point of view, the goal of this review is to discuss the approaches available to investigate ER–mitochondria contact sites, taking into account that every single method can be useful for a specific range of information depending on the topic of the research.

## 2. Proteins Involved in ER–Mitochondria Interplay

Among the organelles interacting with the ER, which include the Golgi apparatus, mitochondria, peroxisomes, and the plasma membrane, mitochondria establish one of the most and well-characterized connection [6]. Electron tomography was used to measure ER–mitochondria proximity and it revealed that the distance between the two organelles is approximately of 10–30 nm [3,8]. This distance is close enough to propose that the ER and the outer mitochondria membrane (OMM) can be tethered together by proteins located on the opposing membranes—indeed, mitochondria-associated ER membranes (MAMs) can be physically separated from the other organelle membranes [17,18,19,20]. We also know that their functional interplay [21] is guaranteed by specific, stabilized contacts [22] that are able to stay tethered to each other even when the organelles are moving along the cytoskeleton [23]. Many proteins have been identified in the MAMs but, for many of them, the biological role is not yet fully understood (Figure 1).

In yeast cells, the ER–mitochondria connection is ensured by a multiprotein complex called the ER–mitochondria encounter structure (ERMES), containing Mdm12, Mdm34, Mdm10, and Mmm1 proteins [24]. This physical tethering allows efficient lipid transport by soluble lipid-carrier proteins such as ceramide-transfer protein (CERT) and oxysterol-binding protein (OSBP) [25]. In mammalian cells, the ER–mitochondria interface is more complex. Indeed, a plethora of proteins are involved in maintaining and modulating this interaction. Mitofusin 2 (MFN2) is one of such players. It is a mitochondrial fusion guanosine-5′-triphosphate (GTP)ases, localized at the outer membrane of mitochondria (OMM) and found in MAMs [26,27,28,29]. MFN2 plays a crucial role in mitochondrial fusion together with Optic Atrophy 1 OPA1, another mitochondrial fusion GTPase located on the inner membrane of mitochondria [30]. It has been reported that during the mitochondrial fusion process, mitochondrial MFN2 assembles homo- or heterodimer complexes with ER-residing MFN2 [28,31]. The interaction between the protein tyrosine phosphatase-interacting protein-51 (PTPIP51), located on the OMM, and an ER-resident protein, the vesicle-associated membrane protein B (VAPB), represents another important mechanism of ER–mitochondria tethering. VAPB and PTPIP51 directly interact, and the modulation of their expression level affects the Ca^2+^ exchange between ER and mitochondria [32] and causes alterations in ER–mitochondria contacts [33,34,35]. Other players that have been found in MAMs are the fission protein 1 homologue (Fis1) and the B cell receptor-associated protein31 (BAP31) [36]. In particular Fis1 is located at the OMM and, by recruiting dynamin-related protein 1 (DRP1), participates in the process of mitochondria fission [37]. BAP31 is a chaperone that takes place in the misfolded protein degradation mechanism as well as in apoptosis and it is located in the ER membrane [38]. The interaction between Fis1 and BAP31 at the MAMs is important to create a platform essential for the recruitment of procaspase 8 and the transfer of the apoptotic signal from mitochondria to ER [36]. The inositol 1,4,5-trisphosphate receptor (IP3R), an ER Ca^2+^ channel, and VDAC1 (voltage-dependent anion channel 1), an OMM protein, are also found to be enriched in the MAMs and via chaperone Grp75 (glucose-regulated protein 75), enabling the close juxtaposition between the two organelles to regulate ER/mitochondria Ca^2+^ transfer [39,40] and mitochondrial Ca^2+^ uptake through the mitochondrial calcium uniporter (MCU) [41,42]. The existence of ER–mitochondria juxtaposition finely fits with the Ca^2+^ transfer ability between the two organelles [8], demonstrating the importance of MAMs in maintaining calcium homeostasis. Interestingly, alteration of the ER–mitochondria interaction leads to disruption of Ca^2+^ transfer between the two organelles, to ER stress [43], and to autophagy [44]. Moreover, mitochondrial Ca^2+^ accumulation, essential for the metabolism of mitochondria and adenosine triphosphate (ATP) production [45], is impaired when MAMs are perturbed [21,46]. On the other side, massive and/or a prolonged accumulation of Ca^2+^ into mitochondria can also lead to the opening of the permeability transition pore (PTP) in the inner mitochondria membrane (IMM), inducing apoptosis. Considering the importance of maintaining proper Ca^2+^ homeostasis, it is not surprising that many additional components act at the MAMs to tightly regulate the ER–mitochondria crosstalk. Notably, the sarco/endoplasmic reticulum (SR/ER) Ca^2+^ ATPase (SERCA), which is crucial to maintaining bulk cytosolic Ca^2+^ concentration at the basal level and replenishing intracellular stores, also exerts an important role in the control of local Ca^2+^ transfer at ER–mitochondria contacts [47]. Different proteins, located on both ER and mitochondria, have been found to regulate/modulate the SERCA activity, such as transmembrane chaperone calnexin (CNX) [48] or the thioredoxin-related transmembrane protein (TMX1) [49] on the ER side, and Rho GTPase (Miro) on the mitochondria side [50]. Furthermore, other proteins can impact on Ca^2+^ transfer, acting directly or indirectly on some component of the IP3R–Grp75–VDAC–MCU axis, such as calreticulin, whose expression has an inhibitory effect on the IP3R [51], while calreticulin-deficient cells have reduced Ca^2+^ storage capacity in the ER and delayed agonist-mediated Ca^2+^ release [52].

## 3. Cellular Functions Associated with ER–Mitochondria Tethering

In addition to Ca^2+^ signaling, other cellular functions have been linked to MAMs. Even though it is out of the scope of this review to provide an in-depth focus on MAM functions and dysfunctions, we report a general overview since we believe that it is important to understand them in order to better appreciate the reports available that investigate the ER–mitochondria interplay. The first functional role proposed for MAMs is related to lipid synthesis and transfer [53]. In fact, although the ER is the leading organelle for the biosynthesis of lipids, some of their synthetic pathways require the concerted action of enzymes that are located on both ER and mitochondria and thus need to be transferred between the two compartments [54,55]. As a consequence, MAMs are enriched in enzymes involved in lipid synthesis and are fundamental in both lipid synthesis and transfer between ER and mitochondria [56,57,58]. However, the molecular mechanisms underlying this lipid exchange, thought to be non-vesicular, are still under investigation in mammals [59], while in yeast both ERMES and the ER membrane protein complex (EMC) have been linked to lipid trafficking [60,61]. MAMs are also involved in controlling the mitochondrial dynamics of fission and fusion. Mitochondrial division events occur at ER–mitochondria contact sites [22], where different regulators [62,63,64] can anchor the cytosolic dynamin-related protein 1 (Drp1) to the OMM in order to mediate the mitochondria constriction. Drp1, recruited at MAMs, through its GTPase activity forms helical oligomers that wrap around mitochondria, inducing the fission event [65]. However, mitochondria fission may happen even when Drp1 is downregulated thanks to the action of the ER-localized inverted formin 2 (INF2), which in addition to being required for Drp1 recruitment can also initiate actin-dependent constriction of mitochondria [66]. INF2-mediated actin polymerization at MAMs stimulates an increase in mitochondrial Ca^2+^ concentration that is required for mitochondrial fission [67]. The mitochondrial fusion process is mediated by Mfn1 and Mfn2 [68]. Notably, Mfn2 localizes not only at the mitochondrial outer membrane but also at ER membranes, and hence at MAMs [69]. Growing evidence is showing a link between MAMs and autophagy process [70]. The essential role of Ca^2+^ homeostasis in controlling the autophagic process is well established. In particular, an impaired Ca^2+^ transfer can negatively affect the ATP production with subsequent increase of cytosolic adenosine monophosphate (AMP)/ATP ratio, which can lead to activation of AMP-activated protein kinase (AMPK) and autophagy [71]. Furthermore, Hamasaki and colleagues demonstrated that, under starvation, different autophagy-related proteins were enriched at ER–mitochondria contact sites, and that an alteration in the crosstalk between ER and mitochondria led to a decrease in autophagy due to defects in autophagosome formation [72]. The ER–mitochondria tethering also plays a role in the inflammation process—it has been demonstrated that ROS production at the ER-mitochondria contact sites [73] can induce the activation and assembly of the inflammasome [74], i.e., a multi-protein complex formed after the activation of nucleotide-binding oligomerization domain (NOD)-like receptor protein 3 (NLRP3) by cellular infection or stress and that triggers the maturation of pro-inflammatory cytokines [75]. The ER–mitochondria tethering is also important during apoptotic signaling. Indeed, apoptosis induction promotes physical contacts between BAP31 and Fis1 that are required for caspase activation and the cleavage of BAP31. The release of the pro-apoptotic p20 fragment, upon binding to the ER membrane, induces ER Ca^2+^ release. Increased Ca^2+^ concentration at the MAMs promotes mitochondrial Ca^2+^ uptake and the recruitment of Drp1, which leads to mitochondrial fragmentation [76]. To sum up, ER–mitochondria interaction emerged as a complex hub fundamental for the integration of numerous cell pathways.

## 4. Experimental Approaches to Study ER–Mitochondria Contact Sites

As reported above, it is clear that the involvement of the interplay between ER and mitochondria in so many physiological processes requires that these interactions should be limited to specific contact site regions and that they can be transient and change in their abundance. All these characteristics make their investigation a big challenge. The detection and the characterization of the contact site as well as the characterization of the resident proteins are essential to properly define organelle membrane proximity as a real contact site. To date, an optimized approach that can provide all these aspects at once is still missing. Hence, a combination of different methods should be employed to obtain a complete description of the organelle contact sites. In the following paragraph, we report an overview of the available approaches suitable for the investigation of ER–mitochondria contact sites. Table 1 briefly summarizes the advantages and disadvantages of these approaches depending on the goal to be achieved and some intrinsic limitations that one should pay attention to when interpreting data.

### 4.1. Fluorescent Probe-Based Methods

Fluorescent proteins (FPs) targeted to the two cellular compartments of interest have been one of the first used methods to identify organelle connection. This approach was introduced to identify ER–mitochondria juxtaposition as site of Ca^2+^ transfer between the two organelles [8] and the expression of targeted green fluorescent protein (GFP) spectral variants into different organelles have been used to visualize multiple organelle contacts at once. However, organelle contacts in confocal images are defined as an overlap of three or more contiguous pixels between segmented features, with the target feature dilated by 1 pixel (equivalent to 97 nm). This may lead to an overestimate of contacts since the distance between ER and mitochondria at membrane contact sites is in the 15–30 nm range [77]. If on one hand this approach can be useful to detect ER–mitochondria interaction and is compatible with live imaging, on the other it does not permit the accurate estimation of the interorganelle distance and the possibility to visualize contact site dynamics.

In an effort to solve these issues, several approaches have been developed. One of these is the dimerization-dependent fluorescent protein (ddFP) technology [78] that is based on the formation of a fluorescent heterodimeric complex upon interaction of two dark FP monomers. Although this interaction shows very low intrinsic affinity, the fluorogenic response due to the FP–FP binding is an indicator of an increase in the proximity or in the effective concentration of monomers. Thus, by targeting each monomer to two subcellular compartments, the inter-organelle proximity can be appreciated as protein fluorescence reconstruction that occurs when the two dark FPs are bound together. The initial ddFP system was composed by a red fluorescent system (ddRFP) derived from dTomato [79], although such first-generation construct suffers from limited brightness and color palette (i.e., only red) [78]. Alford et al. expanded the color palette of ddFPs with variants exhibiting improved brightness and contrast—by a process of directed evolution, the authors created green (ddGFP) and yellow (ddYFP) analogues of ddRFP [80]. An evolution of the protein fragment complementation assay (PCA) based on fluorescent proteins, developed to analyze the spatiotemporal dynamics of protein–protein interactions (PPIs) [81], can be applied to study the ER–mitochondria interaction by fusing the coding sequence of known tethers to the N- and C-terminal fragments of the FP. Both ddFP and PCA are intriguing methods to evaluate ER–mitochondria dynamics since the interaction between the proteins/fragments is reversible or partially reversible. However, their low-fluorescence intensity prevents their application in real scenarios.

Fluorescence resonance energy transfer (FRET) represents another fascinating method for investigating contact site dynamics, being extremely sensitive to the distance between membranes. This approach is based on the detection of energy transfer between two proximity fluorophores targeted to the organelles of interest. Such fluorophores contain the rapamycin-induced dimerization domain [82] and dimerize upon rapamycin addition, giving rise to maximal FRET signal. This approach has been firstly used to visualize the ER–mitochondria juxtaposition [83]; however, in principle, by adding the appropriate targeting sequence to the individual fluorescent proteins, it could be adapted to measure any potential contact site. FRET probe applications were limited by the fact that this approach requires equimolar expression of the two moieties. This aspect has been recently solved by expressing the two fluorophores as a single mRNA with a self-cleavable Thosea asigna virus 2A siteTAV2a sequence between them. However, it has been observed that FRET sensitivity is inversely proportional to the sixth power of the distance between the fluorophores, demonstrating that this approach is highly sensitive to the distance between the two membranes [84] and the requirement of rapamycin [85,86] can limit its application range.

A new generation of contact site sensors based on split-GFP (SPLICS) has been developed to measure ER–mitochondria contact sites over a range of distances and can be easily adapted to other types of hetero- and homotypic contact sites. The split GFP system is a combination of two non- fluorescent portions of the superfolder GFP variant, a GFP fragment containing the β-strands 1–10 and the β-strand 11, which can spontaneously refold to reconstitute the complete β-barrel structure of the fluorescence-emitting GFP [87,88]. Each moiety can be targeted to one of the juxtaposed membranes of interest—the GFP fluorescence will be restored only when the two portions are close enough. Interestingly, the assembled split-GFP signals were observed as discrete foci between different pairs of organelles, i.e., ER, mitochondria, vacuole, peroxisomes, and lipid droplets (LDs), suggesting that each organelle forms contact sites with limited areas of the membrane of different organelles simultaneously [89,90,91]. Thus, the split-GFP system could be used as a potential tool for the screening of unidentified tethering factors between two organelles and/or their regulator proteins. Although the complementation of the two fragments spGFP1-10 and spGFP11 has been reported to be thermodynamically stable in in vitro experimental conditions [87], it is not clear whether the interaction could be made reversible when spGFP1-10 and spGFP11 are fused to two proteins that could be pulled apart under certain conditions in living cells. Cieri and colleagues proposed two split-GFP variants to measure short- (≈8–10nm) and long-range (≈40–50 nm) ER–mitochondria interactions [89]. The two constructs differ in the length of the spacer placed between the ER targeting sequence and the β11 fragment and were created by considering the distance of 0.36 nm between two alpha-carbons in a peptide chain—the ER-Short β11 has a 29 aa spacer and the ER-Long β11 has a 146 aa spacer. Interestingly, this method has been proven to detect ER–mitochondria interplay in vivo in zebrafish sensory neurons [89].

Light microscopy, confocal microscopy in particular, is the most common technique used to visualize all of the fluorescent probes at the ER–mitochondria interface reported above [8]. However, confocal microscopy suffers from an intrinsic limitation due to low spatiotemporal resolution. To overcome this problem, super-resolution fluorescence microscopy (SRM) offers the possibility to increase the high temporal and spatial resolution to better appreciate the dynamics and fine structure of ER–mitochondria contact sites [92,93]. Structured illumination microscopy (SIM), a type of SRM, is suitable for fast live-cell imaging and has been used to reveal subcellular structures and dynamics of ER–mitochondria contacts [94,95]. However, the use of this kind of approach is still limited in the characterization of ER–mitochondria contact sites as it requires highly dedicated microscopes and technical expertise, thus resulting in an expensive and difficult technique.

### 4.2. Immunodetection-Based Methods

ER–mitochondria interaction is mediated by proteins that act as crucial players in maintaining organelle tethering [24,96,97]. For this reason, some approaches based on the immunodetection of proteins can be applied for the identification of the interplay between the two organelles. Interestingly, such methods can be also used to measure organelle distances. For instance, the proximity ligation assay (PLA), which is used to detect the highly specific proximal binding of protein pairs in cells or tissues [98,99], can be adapted for the study of proteins resident at the two-membrane interface. This method is based on the use of proximity probes that are constituted by oligonucleotides attached to antibodies against the two target proteins. The binding of the two proteins can guide the formation of circular DNA strands when they are close enough. These DNA circles serve as templates for the rolling-circle amplification (RCA), covalently linked to an antibody–antigen complex. Final RCA can be detected through the hybridization of the complementary fluorescence-labeled oligonucleotide. By varying the protein binders (antibodies) and the length of the oligonucleotides on the proximity probes, this method could theoretically be used as a molecular ruler, allowing measurements of distances between epitopes. Indeed, Soderberg and colleagues measured that the maximum distance between determinants recognized by PLA is estimated at roughly 30 nm, including the size of the two antibodies and the oligonucleotides connecting them in the detected protein pairs. If required, longer distances can be measured using longer oligonucleotides. More compact binders and shorter DNA sequences can be used to improve resolution by limiting detection distances to just over 10 nm [100]. However, one of the main limitations of this method is that PLA signal is due to PLA partner expression, which can change in total amount. Furthermore, PLA partners are not always unequivocally expressed into the contact sites between two organelles, leading to increased background. Finally, this analysis provides a snapshot of cellular processes because samples must be fixed before the analysis, even though fixation may introduce artifacts.

It is worth mentioning that an engineered version of ascorbate peroxidase (APEX) has been used to identify ER–mitochondria resident proteins [101,102]. Differently from the commonly used horseradish peroxidase (HRP), APEX retains its activity when expressed in the cytosol, mitochondria, and in other reducing environments within the cell. Upon live cell treatment for 1 min with H_2_O_2_ in the presence of biotin-phenol, APEX catalyzes the one-electron oxidation of biotin-phenol to generate a very short-lived biotinphenoxyl radical. This radical covalently biotinylates endogenous proteins proximal to APEX. Employing streptavidin pull-down protocol makes the extraction of biotinylated players easy. Lam and colleagues demonstrated that the introduction of a single A134P substitution in APEX generates a more sensitive and stable peroxidase [101]. Interestingly, this approach can be associated with mass spectrometry and used to identify new resident proteins at the ER–mitochondria interface, as reported by Cho and colleagues [102]. By targeting the APEX probe to the outer surface of the membranes of two organelles, it has been possible to map the entire proteome profile related to specific intra-organelle communications [103]. A split version of APEX2 has also been developed—two inactive fragments of APEX2 reconstitute to give an active peroxidase only when they are physically at close proximity. The inactive fragments of APEX2 are bound to the rapamycin-induced dimerization domain and require the rapamycin addition to reconstitute the APEX native structure [104]. As for the FRET approach, it is easy to understand that the use of rapamycin limits the application of this new APEX system. Figure 2 summarizes the techniques that have been described.

### 4.3. Electron Microscopy-Based Methods

Transmission electron microscopy (TEM) is one of the first methods used to detect ER–mitochondria contact sites [1]. Through a high voltage electron beam illuminating the specimen, this approach can provide high-nanoscale resolution images of contact site architecture within the cellular context. Electron microscopy imaging is useful in revealing the morphological diversity of contact sites, even though only highly abundant ones can be properly detected [3,105,106]. Although TEM provides information on the inner structure of the specimen such as structure morphology, it gives only a two-dimensional (2D) view, neglecting information that can only be appreciated with three-dimensional (3D) methods. To overcome this limitation, a 3D electron tomography (ET) technique has been developed, which enables the reconstruction of the missing third dimension by combining two perpendicular 2D projection tilt series, with a final lateral resolution between 3 and 8 nm. In ET, multiple images are captured as the sample is tilted along an axis. The images are then aligned and merged using computational techniques to reconstruct a 3D picture or tomogram [107,108]. Both TEM and ET can be combined with immunostaining approaches to detect protein localization in the 2D or 3D reconstruction of cells or tissues [107]. Immunogold staining is one of the most suitable techniques used to improve visualization of cellular details since gold probes are the most reliable choice for immunostaining in electron microscopy for their high electron density, biocompatibility, and excellent electrical thermal conductivity [109]. Such a protocol can introduce artifacts due to fixation procedure. To overcome this limitation, an intriguing protocol to generate a reconstruction of cellular structure in a fully hydrated environment has been developed. Biological samples are infiltrated with sucrose followed by cryo-sectioning of the frozen specimen at temperatures between −80 and −120 °C. This technique can preserve both cellular ultrastructure and protein epitopes, enabling powerful immunolocalization studies [110]. Despite the fact that ET can provide more useful information compared to TEM, reliable full 3D reconstructions are not always guaranteed because of limited tilt range of the sample holder that can leave empty regions.

In the range of microscope-based approaches, the scanning EM (SEM) technique has also been proposed to investigate the ER–mitochondria interaction of large specimen volumes. Common SEM uses low electron energies to provide surface information, while backscattered electrons can be used to obtain information from the first few nanometers below the surface, ensuring high Z resolution [105,111]. Over the past 20 years, the development of new methods has led to a higher quality of full 3D image reconstruction, with increased resolution efficiency. These methods use either an automated ultramicrotome located in the SEM chamber to obtain a serial block-face imaging [112] or a focused ion or plasma beam (FIB-SEM) [113] to thinly dissect a hard substrate, also performed in the SEM chamber. Both processes can run in an automated manner to collect many hundreds of serial images. Growing evidence has reported that immunogold staining can also be applied to SEM to detect resident proteins on the ER–mitochondria interface [114]. To sum up, despite these recent improvements, the ability to obtain an appropriate image is still limited since extended research time and computer power are required to process a large amount of data.

### 4.4. Cell Fractionation

Biochemical characterization of ER–mitochondria contact sites can be investigated by performing subcellular fractionation via sucrose gradient centrifugation. In the late 1950s, ER–mitochondria interaction was observed in liver mitochondria preparations obtained from sucrose fractionation; however, it was thought to be due to a contamination [115]. Since then the protocol to enrich the MAM portion during the cell/tissue fractionation has been greatly improved and the association between ER and mitochondria has now been well established through this approach [32,36,39]. Wieckowski and colleagues published a detailed guideline to purify both MAM and pure mitochondria (pure mito) starting from HeLa cells or rat liver tissue. First of all, isolation of crude mitochondria, containing MAM and pure mito, is needed. After that, MAM and pure mitochondrial fraction can be isolated from each other by adding different amounts of mannitol and Percoll-containing solutions, followed by several numbers of high-speed centrifugations in order to discriminate both parts [116]. To validate the proper MAM isolation, some controls are required. In particular, Western blot analysis of specific markers should be performed to confirm and characterize a good fractionation. For positive control, known protein enriched at MAM can be used, among them IP3Rs, fatty acid-CoA ligase 4 (FACL4), and VDAC are commonly used. When possible, to discriminate between ER and pure mitochondria contaminations, proteins that are present exclusively in these two organelles and not in MAM should be checked; among them, cytochrome-c and NADH dehydrogenase 1 alpha subcomplex subunit 9 (NDUFA9) for mitochondrial and calnexin or calreticulin for MAM/ER marker are commonly used. Finally, the absence of other organelle contaminations, i.e., lysosomes, Golgi apparatus, peroxisomes, and nuclear and plasma membranes should also be verified. To date, cell fractionation remains one of the most commonly used techniques to investigate the ER–mitochondria tethering, leading to the possibility to discover new players resident at MAM since, after MAM purification, proteomic analysis can be performed to identify new MAM resident proteins [117]. However, it must be taken into account that the long procedure of this purification technique can alter resident proteins since modifications and interactions (such as phosphorylation or dimerization) can be introduced and that the isolation of pure contact site resident proteins can be compromised by contaminations by other membranes. For these reasons, their identification should be validated by other approaches.

## 5. Discussion

As reported in the second paragraph, the ER–mitochondria tethering is characterized by four main players, IP3R–GRP75–VDAC [39], BAP31–FIS1 [36], Mfn2 [28], and VAP–PTPIP51 [32]. However, to date, more proteins have been reported to be resident at MAMs [118,119]. Indeed, although cell fractionation and EM are the primary and mostly commonly used techniques amenable to discover and investigate ER–mitochondria tethering [39,120], BAP31–FIS1 [36], VAP–PTPIP51 [32], a deeper investigation has been reached in terms of the development of ameliorate approaches. For example, FRET ad ddFP approaches were applied to characterize Mfn2 tether in MEF cells [84]. Furthermore, Gomez-Suaga and colleagues used the PLA to identify the role of the VAPB–PTPIP51 complex in regulating autophagy [34]. A new generation of contact site sensors based on split-GFP have been used to demonstrate that the ER–mitochondria interaction is a dynamic structure that undergoes active remodeling under different cellular needs [91], while Cieri and co-workers reported the possibility of overcoming the light microscopy resolution limits using proximity sensors [89]. Interestingly, APEX represents a new intriguing method to focus on ER–mitochondria contacts in discovering new players [103]. Finally, the improvement on microscopy was applied to meticulously visualize the ER–mitochondria structure in COS-7 (CV-1 in Origin with SV40 genes) and U2OS (Human Bone Osteosarcoma Epithelial Cells) cells [95]. Taking into account the considerations above, it is evident that the study of ER–mitochondria interaction is an extensive field and thus a deep investigation is necessary to thoroughly characterize the interplay. For this reason, we believe that a series of minimal and essential information is required to define whether ER and mitochondria form a contact site. This review outlines that the use of just one of the available methods is not enough to provide all the details, and that a complete, optimized approach to do so is still missing. Therefore, the choice of a technique depends on the type of readout that is expected to be obtained (Figure 3). We believe that all the fluorescence probe-based methods as well as TEM, ET, and SEM are the most accurate approaches to detect the direct interaction between two proximal membranes. The interaction can be also evaluated by PLA whenever the resident proteins are known. Conversely, cell fractionation and APEX are considered unconventional methods and more useful for the isolation and biochemical characterization of resident proteins at contact sites. To gather insights on structure characterization, such as the spatiotemporal dynamics of the contact site, ddFP technologies, PCA based on fluorescence proteins, FRET probes, and the split-GFP-based contact site sensors can be employed, since the reported methods are based on the interaction of two different fragments in a close distance. Differently, the quantification of the distance between the two membranes can be assessed by FRET probes, SPLICS, PLA (if the resident proteins are known), and electron microscopy-based approaches. It is rather obvious that the morphologic structure of the contact site between the **membranes** of two organelles can be visualized only through TEM, obtaining an accurate 2D reconstruction, or by electron tomography (ET). Interestingly, SEM methodologies seem to be useful for 3D reconstruction of specimens with larger volumes. Finally, the biochemical characterization of resident proteins at the contact site is essential to define players at the membrane interface as well as their activity, providing information on possible physiological functions of the contact site itself. This goal can be reached by employing the cell fractionation technique or the APEX approach. Moreover, they can be combined with proteomic and mass spectroscopy analysis to discover new players.

## 6. Conclusions

The study of ER–mitochondria contact sites, as well as of other organelle membranes, is evidently very challenging but also high demanding since it is evident that the available approaches need to be combined to obtain a detailed picture of the nature of the ER-mitochondria contact sites. The identification of the players involved is very important in order to develop drugs that can modulate the contact sites—to this end, a multi-disciplinary integrated approach that combines fluorescence, EM, and biochemistry studies (immunodetection and cell fractionation) is essential.

## Figures and Tables

**Figure 1 ijms-21-08157-f001:**
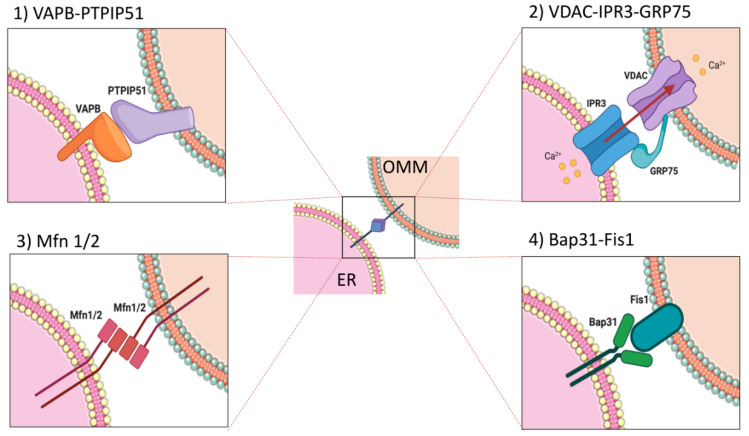
Schematic representation of the tethering complexes at the endoplasmic reticulum (ER)–mitochondria interface.

**Figure 2 ijms-21-08157-f002:**
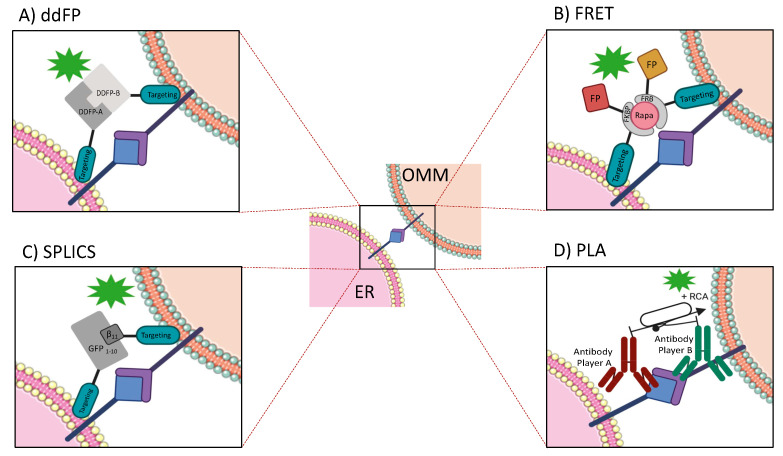
Representation of methods useful to detect proximity interaction between the ER and the outer mitochondria membrane (OMM). The diagram in the center is a schematic interaction of contact site resident proteins at the ER and OMM interface. (**A**) Dimerization-dependent fluorescent proteins (ddFP). The interaction of two dark fluorescent protein (FP) monomers, targeted on organelle interface, recreates a detectable fluorescent heterodimeric complex. (**B**) Fluorescence resonance energy transfer (FRET). Detection of energy transfer between two proximity fluorophores targeted to organelles of interest after rapamycin addition. Fluorophores consist of organelle targeting sequences, rapamycin-induced dimerization domains, and fluorescent proteins. (**C**) Split-GFP-based contact site sensor (SPLICS). The interaction between two proximity non-fluorescent portions of the superfolder GFP variant, the β-strands 1–10, and the β-strand 11 of the GFP spontaneously reconstitutes the complete β-barrel structure of the fluorescence-emitting GFP. (**D**) Proximity ligation assay (PLA). Two antibodies against proteins of interest are attached to oligonucleotides that guide the formation of circular DNA strands when proteins are close enough. The addition of rolling-circle amplification (RCA) permits the detection of protein interaction.

**Figure 3 ijms-21-08157-f003:**
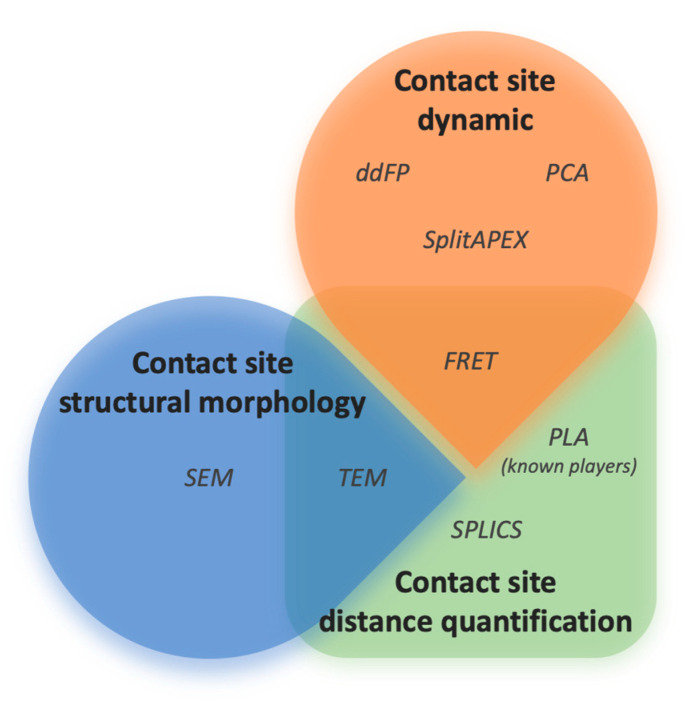
Three main sub-groups of useful methods to characterize ER–mitochondria contact sites have been identified, which provide information on contact site dynamics, contact site distance, and contact site structural morphology. Methods are clustered on the basis of their utility to output a specific information; at intersections, those amenable approaches able to provide more than one information on contact sites have been reported. Abbreviations: ddFP (dimerization-dependent fluorescent protein) [79], PCA (protein-fragment complementation assay) [81], SplitAPEX (split-ascorbate peroxidase) [104], PLA (proximity ligation assay) [100], SPLICS (split-GFP) [89], TEM (transmission electron microscopy) [105], and SEM (scanning electron microscopy) [111].

**Table 1 ijms-21-08157-t001:** Methods to assess ER–mitochondria contact sites and their advantages/disadvantages.

	Method	Advantages	Disadvantages	Limitations
**Fluorescent probe-based**	Old FP	Fast to detect contact sites Live compatible Easy to perform Cheap	Overestimation of contacts sites distance Fixation for immunofluorescence can introduce artifacts	Use of genetically encoded probes Resolution limits of common microscopies used to detect the probes signal Lacking structural information
	ddFP and PCA	Easy to detect contact sites Live compatible Useful to detect contact sites dynamics	Not suitable for distance measurement Low fluorescence of probes Fixation for immunofluorescence can introduce artifacts	
	FRET	Easy to detect contacts sites Live compatible Sensitive to organelle distances Useful to detect contact sites dynamics	Requires equimolar expression of the two moieties Rapamycin addition Fixation for immunofluorescence can introduce artifacts	
	SPLICS	Easy to detect contact sites Live compatible Extremely sensitive to organelle distances Partially useful to detect contact sites dynamics No rapamycin addition	Requires equimolar expression of the two moieties Could be thermodynamically stable Fixation for immunofluorescence can introduce artifacts	
**Immunodetection probe-based**	PLA	Easy to detect contact sites (when players are known) Extremely sensitive to organelle distances	Fixation for immunofluorescence can introduce artifacts Requires antibodies to the proteins of interest PLA partners are not always unequivocally expressed at the contact sites	Availability of specific antibodies Indirect measurements of contact sites due to required chemical reaction to detect the players
	APEX	Biochemical characterization of players at contact sites Combined with proteomic can be used to discover new resident proteins Samples are not contaminated by other organelles	Does not allow to measure the distance at contact sites Does not provide information on spatiotemporal dynamics except for SplitAPEX Fixation can introduce artifacts	
**Microscope-based**	TEM	Morphology structure of contact site within the cells in 2D reconstruction or 3D with ET Can be combined with immunostaining to localize resident proteins at contact sites Measurement of contact sites distances	Fixation can introduce artifacts Does not provide information on spatiotemporal dynamics Useful only for highly abundant contact sites In ET the full 3D reconstructions are not always obtained due to limited tilt range of the sample holder	Information on functionality of contact sites are missing Use of antibodies to detect resident proteins
	SEM	Better quality for morphology structure of contact sites in 3D reconstruction of large specimen volumes Can be combined with immunostaining to localize resident proteins at contact sites	Fixation can introduce artifacts Big challenging Time-consuming and intensive computational processing of data Does not provide information on spatiotemporal dynamics Expensive approach	
**Biochemical**	Cell Fractionation	Biochemical characterization of players at contact sites Combined with proteomic can be useful to discover new resident proteins	Long procedure can introduce biochemical modification altering resident proteins at contact sites Difficulty to isolate pure contact site as contamination are common	Information on quantification, structure and functions of contact sites are not provided Appropriate markers to check other organelles contaminants

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
