# Peer review of "ER–Mitochondria Contact Sites Reporters: Strengths and Weaknesses of the Available Approaches"

_ijms, 2020, doi:10.3390/ijms21218157_

Round 1
Reviewer 1 Report
In this manuscript, Giamogante et al. describe the methods to analyze ER-mitochondria contact sites. This a recommendable theme and the methods and, most importantly, their limitations are discussed.
However, I have some important remarks:
- The section "4.4 Cell fractionation" needs to be extended. This 'isolation of MAMs' is very frequently used in literature to show that proteins are present in these contact sites. However, in many of these studies, there are problems with the negative controls: rarely do I see the validation of the absence of a ER or mitochondria protein that is not in the MAMs. In other words, it appears to me that the techniques to isolate MAMs are not restricted to these contact sites, but isolate general ER and mitochondria membranes. This is probably what the authors refer to as 'contaminations by other membranes'. Therefore, I feel that this section could strongly benefit from a more comprehensive evaluation of this particular technique and its caveats. What is the evidence that this isolation leads to 100% pure MAMs?
- In that sense, please evaluate the evidence that the proteins described in section 2 are indeed present in the MAMs. Has this been described and confirmed independently? Have they used the proper controls?
- I would recommend to include how one should interpret data in the literature with caution regarding analysis of the MAMs. This could be included in Table 1: next to the advantages and disadvantages, describe the issues the reader should point attention to when coming across these techniques in a publication. This could be very helpful.
- Regarding the involvement of the MAMs in apoptosis and autophagy (described in section 3), the most likely regulation would probably go through Ca2+ signaling between the 2 organelles, instead of the mechanisms described. Enhanced Ca2+ signals could lead to mitochondrial overload (apoptosis), while reduced Ca2+ would lead to less ATP and therefore autophagy induction through AMPK (see e.g. PMID 21146562). Regarding the latter, please aslo adapt in section 2 so that ATP regulation by Ca2+ is described first and then link it to autophagy.
- Considering the fluorescent probe-based methods: what about superresolution microscopic techniques?
- Please define the abbreviations of the figure 1 in the legend.
- Line 135-136: "MAMs are consistently enriched in enzymes catalyzing lipid synthesis": please revise.
Author Response
Reviewer 1
In this manuscript, Giamogante et al. describe the methods to analyze ER-mitochondria contact sites. This a recommendable theme and the methods and, most importantly, their limitations are discussed.
However, I have some important remarks:
Response:
We thank the reviewer for considering the manuscript dealing with a recommendable theme whose methods and limitations are described.
The section "4.4 Cell fractionation" needs to be extended. This 'isolation of MAMs' is very frequently used in literature to show that proteins are present in these contact sites. However, in many of these studies, there are problems with the negative controls: rarely do I see the validation of the absence of a ER or mitochondria protein that is not in the MAMs. In other words, it appears to me that the techniques to isolate MAMs are not restricted to these contact sites, but isolate general ER and mitochondria membranes. This is probably what the authors refer to as 'contaminations by other membranes'. Therefore, I feel that this section could strongly benefit from a more comprehensive evaluation of this particular technique and its caveats. What is the evidence that this isolation leads to 100% pure MAMs?
Response:
We thank the reviewer for the important comment. We agree with this point and we have deeply modified the section by adding the following paragraph: “Since then the protocol to enrich the MAM portion during the cell/tissue fractionation has been greatly improved and the association between ER and mitochondria has now been well established through this approach [119-121]. Wieckowski and colleagues published a detailed guideline to purify both MAM and pure mitochondria (pure mito) starting from Hela cells or rat liver tissue. First of all, isolation of crude mitochondria, containing MAM and pure mito, is needed. After that, MAM and pure mitochondrial fraction can be isolated from each other by adding different amounts of mannitol and Percoll-containing solutions, following by several numbers of high-speed centrifugations in order to discriminate both parts [116]. To validate the proper MAM isolation some controls are required. In particular, western blot analysis of specific markers should be performed to confirm and characterize a good fractionation. For positive control, known protein enriched at MAM can be used, among them IP3Rs, FACL4 and VDAC are commonly used. When possible, to discriminate between ER and pure mitochondria contaminations, proteins present exclusively in these two organelles, and not in MAM, should be checked, among them Cytochrome-c and NDUFA9 for mitochondrial and Calnexin or Calreticulin for MAM/ER marker are commonly used. Finally, the absence of other organelles contaminations, i.e., lysosomes, Golgi apparatus, peroxisomes, nuclear and plasma membranes should be also verified. To date, cell fractionation remains one of the most used techniques to investigate the ER-mitochondria tethering leading to the possibility to discover new players resident at MAM since, after MAM purification, proteomic analysis can be performed to identify new MAMs resident proteins [117].”
In that sense, please evaluate the evidence that the proteins described in section 2 are indeed present in the MAMs. Has this been described and confirmed independently? Have they used the proper controls?
Response:
We thank the reviewer for the comment. Indeed all the players cited in the text have been validated, confirmed and described as MAM resident proteins by cell fractionation and microscopy procedures in the references that we have cited in the text ( i.e., Filadi R, et al. Proc Natl Acad Sci U S A. 2015 Apr 28;112(17):E2174-81; de Brito, O. M.; Scorrano, L., Nature 2008, 456, (7222), 605-10; De Vos, K. J et al. Hum Mol Genet 2012, 21, (6), 1299-311; Iwasawa, R.et al. EMBO J 2011, 30, (3), 556-68; Szabadkai, G. et al. J Cell Biol 2006, 175, (6), 901-11). The demonstration of the localization at MAMs of the above cited players is out of the scope of the present contribution, In most of the cases positive and negative controls are present and the purification procedure includes enrichment of the MAM compartment by ultracentrifugation where some players are found and many others are not. This is in our opinion a convincing evidence that the proteins claimed to be resident at MAMs are genuine MAM-located.
I would recommend to include how one should interpret data in the literature with caution regarding analysis of the MAMs. This could be included in Table 1: next to the advantages and disadvantages, describe the issues the reader should point attention to when coming across these techniques in a publication. This could be very helpful.
Response:
We thank the reviewer for raising this important point. We do also agree with the fact that this would greatly improve the comprehension of the manuscript. We have now changed the table and added a new section reporting the Limitations of each the techniques used.
Regarding the involvement of the MAMs in apoptosis and autophagy (described in section 3), the most likely regulation would probably go through Ca2+ signaling between the 2 organelles, instead of the mechanisms described. Enhanced Ca2+ signals could lead to mitochondrial overload (apoptosis), while reduced Ca2+ would lead to less ATP and therefore autophagy induction through AMPK (see e.g. PMID 21146562). Regarding the latter, please aslo adapt in section 2 so that ATP regulation by Ca2+ is described first and then link it to autophagy.
Response:
Thank you for the point. We have added the requested reference and changed section 3 by adding the following statement: “It is well established the essential role of Ca2+ homeostasis in controlling the autophagic process. Particularly, an impaired ER/mitochondria Ca2+ transfer can negatively affect the ATP production with subsequently increase of cytosolic AMP/ATP ratio, which can lead to activation of AMP-activated protein kinase (AMPK) and autophagy [71]”. Since reviewer 3 also asked to revise this section we have changed its title and made a couple of additional improvements (see lines 154-156).
Considering the fluorescent probe-based methods: what about superresolution microscopic techniques?
Response:
Thank you for the comment. We have now included a short section dealing with the super resolution microscopy technique, however, considering that this approach for the analysis of the MAM is not widely used, the section is rather limited. In particular, at the end of section 4.1 we have added the following statement: “Light microscopy, in particular confocal microscopy, is the most common technique used to visualize all of the above reported fluorescent probes at the ER-mitochondria interface [8]. However, confocal microscopy suffers from an intrinsic limitation due to low spatiotemporal resolution. To overcome this problem, super-resolution fluorescence microscopy (SRM) offers the possibility to increase the high temporal and spatial resolution to better appreciate the dynamics and fine structure of ER-mitochondria contact sites [92, 93]. Structured illumination microscopy (SIM), a type of SRM, is suitable for fast live-cell imaging and has been used to reveal subcellular structures and dynamics of ER-mitochondria contacts [94, 95]. However, the employ of these approaches is still limited in the characterization of ER-mitochondria contact sites since they require highly dedicated microscopes and technical expertise, then resulting in an expensive and difficult technique.”
Please define the abbreviations of the figure 1 in the legend.
Response:
Thank you, fixed.
Line 135-136: "MAMs are consistently enriched in enzymes catalyzing lipid synthesis": please revise.
Response:
Thank you, fixed
Reviewer 2 Report
Giamogante and collaborators summarized the current understanding of mitochondria-ER contacts as well as techniques available to characterize them. This is a comprehensive and timely review, but there are some deficiencies in details. Consequently, here are some suggestions and minor comments.
- Line 28: avoid the expression “do not just fly alone” – can be modified with “are not isolated”.
- Some statements require further in-depth discussion and proper referencing. For example, the first paragraph is too vague and does not add to the review. It should be more specific or brief so that can be joined with the second paragraph.
- Line 56 – Line 65: The three pieces described by the authors as needed to determine whether ER-mitochondria interaction forms are not informative at the beginning of the review. They are weak and lack appropriate references. This section could be complemented or may not be needed.
- The section on proteins involved in the ER-mitochondria interplay is good. A figure accompanying this section could be included to better represent the protein network involved.
- The section on ER-mitochondria interplay in health and disease lacks depth and details about specific original studies linking dysfunctional MAMs to cancer, neurodegeneration, aging, and diabetes. A table could help the reader understand better the state of the field.
- Line 180: “Each method has advantages and disadvantages depending on the goal to be achieved.”: Cite Table 1.
- Figure 1 should include appropriate references.
- Figure 2 is useful but very hard to read. The central portion could be decreased in size, while the specific panels could be increased – including font size.
- The experimental approaches section is useful to describe the different techniques available. It would be useful to have specific examples of how these techniques have been used to study and to characterize MAMs.
Author Response
Reviewer 2
Giamogante and collaborators summarized the current understanding of mitochondria-ER contacts as well as techniques available to characterize them. This is a comprehensive and timely review, but there are some deficiencies in details. Consequently, here are some suggestions and minor comments.
Response:
We are happy to see that also this reviewer considers this contribution comprehensive and timely.
- Line 28: avoid the expression “do not just fly alone” – can be modified with “are not isolated”.
Response:
Thank you, fixed.
- Some statements require further in-depth discussion and proper referencing. For example, the first paragraph is too vague and does not add to the review. It should be more specific or brief so that can be joined with the second paragraph.
- Line 56 – Line 65: The three pieces described by the authors as needed to determine whether ER-mitochondria interaction forms are not informative at the beginning of the review. They are weak and lack appropriate references. This section could be complemented or may not be needed.
Response:
Thank you for the comment. We have changed the introduction accordingly. The following changes have been added: Lines 29-32 “Proteins identified at the interface between organelles seems to act as tethers that physically bridge two opposing membranes and join them one to each other [2, 3]; others have a specific role occurring at the contact sites, such as the ability to transfer small molecules in a non-vesicular manner [4, 5].”
Lines 44-55 “However, most of the research aimed to characterize ER-mitochondria contact-site components and their functions faces the complex issue of unequivocally identifying the contact, defining its function and monitoring its changes upon perturbations. To date, several methods need to be combined to extract all the mentioned information since an optimized approach that can provide them at once is still missing. Indeed, the transient nature and variable abundance of ER-mitochondria interactions within different cell types make the development of suitable detecting methods a big challenge. During the past decades, considerable effort has been spent to ameliorate the available methods and develop new ones, leading to the possibility of further investigate on ER-mitochondria contact sites and provide more information. Starting from this point of view, the goal of this review is to discuss the approaches available to investigate ER-mitochondria contact sites, taking into account that every single method can be useful for a specific range of information depending on the topic of the research.”
In agreement with the comment we have also eliminated the three pieces described.
-The section on proteins involved in the ER-mitochondria interplay is good. A figure accompanying this section could be included to better represent the protein network involved.
Response:
We do agree with the comment. We have now included a new figure accompanying this section. This is shown as new Figure 1
- The section on ER-mitochondria interplay in health and disease lacks depth and details about specific original studies linking dysfunctional MAMs to cancer, neurodegeneration, aging, and diabetes. A table could help the reader understand better the state of the field.
Response:
Thank you for the comment. Indeed, the section itself was more focused on the cellular functions associated with the MAMs while the disease-related part was just cited at the end of the paragraph. Our aim was to let it focused most prominently on the first rather than on the latter therefore we have changed the title of the section: “Cellular functions associated with ER-mitochondria tethering” and deleted the last sentence.
- Line 180: “Each method has advantages and disadvantages depending on the goal to be achieved.”: Cite Table 1.
Response:
Thank you, fixed.
- Figure 1 should include appropriate references.
Response:
Thank you, fixed.
- Figure 2 is useful but very hard to read. The central portion could be decreased in size, while the specific panels could be increased – including font size.
Response:
Thank you, fixed.
- The experimental approaches section is useful to describe the different techniques available. It would be useful to have specific examples of how these techniques have been used to study and to characterize MAMs.
Response:
Thank you for the comment. The discussion section has been modified by adding some examples of how these techniques have been used to study and characterize MAMs. The following statement has now been added: “As reported in the second paragraph, the ER-mitochondria tethering is characterized by four main players, IP3R-GRP75-VDAC [39], BAP31-FIS1 [36], Mfn2 [28] and VAP-PTPIP51 [32]. However, to date more proteins have been reported to resident at MAMs [118, 119]. Indeed, although the cell fractionation and EM are the primary and mostly used techniques amenable to discover and investigate on ER-mitochondria tethering [39, 120], BAP31-FIS1 [36], VAP-PTPIP51 [32], a deeper investigation has been reached with the development of ameliorate approaches. For example, FRET ad ddFP approaches were applied to characterize Mfn2 tether in MEF cells [84]. Furthermore, Gomez-Suaga and colleagues used the PLA to identify the role of VAPB-PTPIP51 complex in regulating autophagy [34]. A new generation of contact site sensors based on split-GFP have been used to demonstrate that ER-mitochondria interaction is a dynamic structure which undergoes active remodeling under different cellular needs [91] while Cieri and co-workers reported the possibility to overcome the light microscopy resolution limits using proximity sensors [89]. Interestingly, APEX represents a new intriguing method to focus on ER-mitochondria contacts in discovering new players [103]. Finally, the improvement on microscopy was applied to meticulously visualize the ER-mitochondria structure in COS-7 and U2OS cells [95]. Taking into account the above considerations,”.
Additionally, table 1 has also been modified in order to take into account the limitations of the different techniques used for the study of the MAMs
Reviewer 3 Report
The review by Giamogante et al will provide a useful resource, placing the importance of the ER-mitochondrial contact in a historical and biological context; and summarizes some of the current experimental approaches to study the cellular domain.
The review (text and figures) focuses on the use tag/genetic approaches to study ER-mitochondrial contacts. Indeed this is a strength of the review. However, discussions regarding biochemical are somewhat superficial and underdeveloped - especially within the context of being an "essential piece" (as emphasized in the introduction & discussion). The majority of experiments identifying novel players in the study of ER-mitochondrial proteins rely heavily upon cellular fractionation methods; increasingly upon proteomics and related biochemical techniques.
The review is well written.
Very Minor:
Introduction. Slang: "just fly alone" seems somewhat out of place.
typo line 206
Author Response
Reviewer 3
The review by Giamogante et al will provide a useful resource, placing the importance of the ER-mitochondrial contact in a historical and biological context; and summarizes some of the current experimental approaches to study the cellular domain.
The review (text and figures) focuses on the use tag/genetic approaches to study ER-mitochondrial contacts. Indeed this is a strength of the review. However, discussions regarding biochemical are somewhat superficial and underdeveloped - especially within the context of being an "essential piece" (as emphasized in the introduction & discussion). The majority of experiments identifying novel players in the study of ER-mitochondrial proteins rely heavily upon cellular fractionation methods; increasingly upon proteomics and related biochemical techniques.
The review is well written.
Very Minor:
Introduction. Slang: "just fly alone" seems somewhat out of place.
typo line 206
Response:
Thank you for considering the contribution a useful resource for the ER-mito contact sites and for considering a well written review. We thank the reviewer for the constructive suggestions that we have carefully taken into account. We have therefore expanded section 4.4 on cell fractionation by adding the following statement: “Since then the protocol to enrich the MAM portion during the cell/tissue fractionation has been greatly improved and the association between ER and mitochondria has now been well established through this approach [119-121]. Wieckowski and colleagues published a detailed guideline to purify both MAM and pure mitochondria (pure mito) starting from Hela cells or rat liver tissue. First of all, isolation of crude mitochondria, containing MAM and pure mito, is needed. After that, MAM and pure mitochondrial fraction can be isolated from each other by adding different amounts of mannitol and Percoll-containing solutions, following by several numbers of high-speed centrifugations in order to discriminate both parts [116]. To validate the proper MAM isolation some controls are required. In particular, western blot analysis of specific markers should be performed to confirm and characterize a good fractionation. For positive control, known protein enriched at MAM can be used, among them IP3Rs, FACL4 and VDAC are commonly used. When possible, to discriminate between ER and pure mitochondria contaminations, proteins present exclusively in these two organelles, and not in MAM, should be checked, among them Cytochrome-c and NDUFA9 for mitochondrial and Calnexin or Calreticulin for MAM/ER marker are commonly used. Finally, the absence of other organelles contaminations, i.e., lysosomes, Golgi apparatus, peroxisomes, nuclear and plasma membranes should be also verified. To date, cell fractionation remains one of the most used techniques to investigate the ER-mitochondria tethering leading to the possibility to discover new players resident at MAM since, after MAM purification, proteomic analysis can be performed to identify new MAMs resident proteins [117].”
We have also fixed the very minor points.
Round 2
Reviewer 1 Report
The authors have addressed the comments in an appropriate manner.
Reviewer 2 Report
The authors have efficiently addressed all of the suggestions and concerns.